# Cryo-electron microscopy reveals two distinct type IV pili assembled by the same bacterium

Alexander Neuhaus[1,2], Muniyandi Selvaraj [3,7], Ralf Salzer[4,8], Julian D. Langer[5,6], Kerstin Kruse[4], Lennart Kirchner[4], Kelly Sanders[1,2], Bertram Daum [1,2], Beate Averhoff[4] & Vicki A. M. Gold [1,2✉]

Type IV pili are flexible filaments on the surface of bacteria, consisting of a helical assembly of pilin proteins. They are involved in bacterial motility (twitching), surface adhesion, biofilm formation and DNA uptake (natural transformation). Here, we use cryo-electron microscopy and mass spectrometry to show that the bacterium *Thermus thermophilus* produces two forms of type IV pilus ('wide' and 'narrow'), differing in structure and protein composition. Wide pili are composed of the major pilin PilA4, while narrow pili are composed of a so-far uncharacterized pilin which we name PilA5. Functional experiments indicate that PilA4 is required for natural transformation, while PilA5 is important for twitching motility.

[1] Living Systems Institute, University of Exeter, Stocker Road, Exeter EX4 4QD, UK. [2] College of Life and Environmental Sciences, Geoffrey Pope, University of Exeter, Stocker Road, Exeter EX4 4QD, UK. [3] Department of Structural Biology, Max Planck Institute of Biophysics, Max-von-Laue Str. 3, 60438 Frankfurt am Main, Germany. [4] Molecular Microbiology and Bioenergetics, Institute of Molecular Biosciences, Goethe University Frankfurt, Max-von-Laue Str. 9, 60438 Frankfurt am Main, Germany. [5] Department of Molecular Membrane Biology, Max Planck Institute of Biophysics, Max-von-Laue Str. 3, 60438 Frankfurt am Main, Germany. [6] Proteomics, Max Planck Institute for Brain Research, Max-von-Laue Str. 4, 60438 Frankfurt am Main, Germany. [7] Present address: Laboratory of Structural Biology, Helsinki Institute of Life Science, 00014 University of Helsinki, Helsinki, Finland. [8] Present address: Structural Studies Division, Medical Research Council—Laboratory of Molecular Biology, Cambridge Biomedical Campus, Francis Crick Ave, Cambridge CB2 0QH, UK. ✉email: v.a.m. gold@exeter.ac.uk

Type IV pili (T4P) are flexible extracellular protein filaments found on many bacteria. They form multifunctional fibres involved in twitching motility, adhesion, immune evasion, bacteriophage infection, virulence and colony formation. T4P have also been linked to DNA uptake, called natural transformation, which is a powerful mechanism that enables genetic adaptation[1–3]. The filaments are homopolymers composed of thousands of pilin subunits, which form helical arrays measuring several micrometres in length. The link between T4P mediated motility and natural transformation is so far unclear.

Depending on the bacterial species, pilins range from 90 to 250 amino acids in length. They are produced as prepilins with a typical class III signal peptide[4,5]. The preprotein is translocated via the Sec pathway into the cell membrane where the signal peptide is cleaved by prepilin peptidase, priming the pilin for incorporation into the growing pilus. Filament assembly is ATP-dependent and occurs at an inner membrane platform which contains PilC, PilM, PilN and PilO[6]. In *Thermus thermophilus*, assembly of pilins into a T4P filament depends on the assembly ATPase PilF, which interacts with the inner membrane platform via PilM[7]. Two retraction ATPases, PilT1 and PilT2, are essential for T4P depolymerisation[8,9]. T4P are extruded through the outer membrane secretin PilQ[10–13]. Recently, it has been suggested that expression of the *T. thermophilus* major pilin PilA4 is temperature dependent, leading to hyperpiliation at suboptimal growth temperatures[14]. The first two in situ structures of T4P assembly machineries were solved only recently in both open (pilus assembled) and closed (pilus retracted) states[11,15], yet detailed information regarding the molecular interactions governing filament assembly was lacking.

Crystal structures of full length pilins or head domains from various bacteria are available in different oligomeric states[6,16–22]. Pilins have a conserved N-terminal α-helix, with a 4–5 stranded antiparallel β-sheet at the C-terminus. The α-helix forms the core of the filament, while the globular β-sheet head domain is solvent exposed and subject to post-translational modification[16,17]. To date, five low-resolution cryo-electron microscopy (cryoEM) structures of isolated T4P have been reported. The first, a 12.5 Å structure from *Neisseria gonorrhoeae*, was sufficient to place crystal structures of pilins into the data but not to resolve their structure within the map[17]. Four subsequent structures from *Pseudomonas aeruginosa*, two *Neisseria* species and enterohemorrhagic *Escherichia coli* have been determined in the 5-8 Å resolution range[23–25].

In this study, we combine different modes of cryoEM (cryo-electron tomography (cryoET) and single-particle cryoEM) with functional data to investigate the T4P of *Thermus*. *T. thermophilus* is a well-established model organism used to study the structure and function of T4P and the natural transformation machinery[3]. Surprisingly, we detect two forms of T4P, a wider and a narrower form. We determine structures of the two filaments at the highest resolution to date (3.2 Å and 3.5 Å, respectively), enabling us to visualise near atomic-level detail and build atomic models for each filament de novo. Our data unambiguously demonstrate that the wider pilus is composed of the major pilin PilA4. Proteomics and knock-out mutants reveal that the narrow pilus consists of a previously unknown pilin, which we name PilA5. Functional experiments reveal that PilA4 is required for the assembly of both types of pilus. We confirm that filaments comprised of PilA4 are involved in natural transformation[26] and filaments comprised of PilA5 are essential for twitching motility. Our results further our understanding of bacterial motility and gene transfer, and will help to guide the development of new drugs to fight microbial pathogens.

## Results

**Thermus assembles two types of pilus**. Cells of *T. thermophilus* strain HB27 assemble T4P pili on their surface[27], predominantly at the cell poles[11]. Performing cryoET on cells grown at the optimal growth temperature of 68 °C revealed two types of pilus, with differences in their diameter. Both filaments are observed to emerge from a large protein channel projecting into the periplasm (Fig. 1a, b). As there is only one secretin (PilQ) and one assembly platform (PilM, PilN and PilO) encoded on the *Thermus* genome[27], this suggests that both filaments are T4P, and that they are extruded through the same core machinery. This is supported by previous studies showing that mutants defective in PilQ cannot extrude any type of filament[27] and that a mutant defective in the PilF assembly ATPase is non-piliated[9], also at low temperature[28]. We have shown previously that transcription of the major pilin gene, *pilA4*, is upregulated at low temperature[14]. To address the question of whether the growth temperature affects pilus assembly, we analysed cells grown at the suboptimal growth temperature of 58 °C by cryoET. Again, two types of pilus were observed (Fig. 1c, d). Pili emerge from T4P complexes only sporadically[11], therefore filaments were isolated from cells in order to investigate their structure in more detail. Both wide and narrow forms of the filament were detected in these preparations (Supplementary Fig. 1a, b).

To investigate the composition of the two pilus forms, we performed a quantitative bottom-up proteomics analysis. Protein abundance was evaluated by Label-Free Quantitation (LFQ) to determine relative enrichment or loss of particular proteins at either 68 °C or 58 °C. At 58 °C, the major pilin subunit PilA4 was identified as the most abundant protein component (Supplementary Fig. 2a). The amount of PilA4 was higherat 58 °C compared to 68 °C, in line with the hyperpiliation phenotype[14]. The second

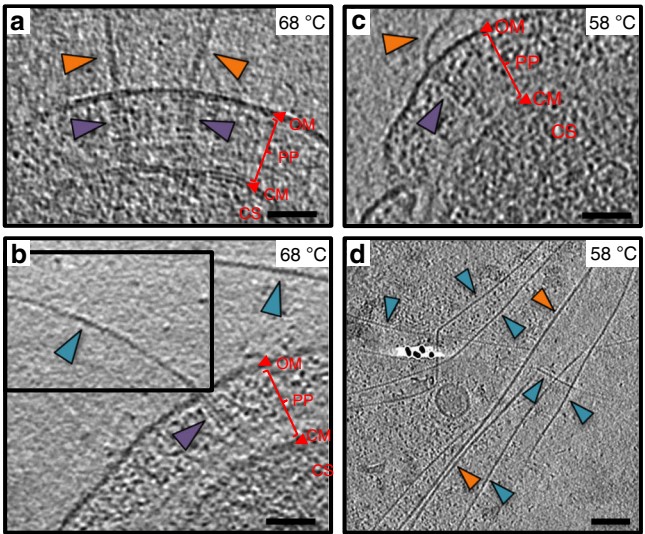

**Fig. 1 Thermus assembles two types of pilus. a**, **b** Tomographic slices through *T. thermophilus* cells grown at 68 °C show both wide (orange arrowheads) and narrow (teal arrowheads) pili emerging from the T4P machinery (purple arrowheads). The pilus emerges from the cell at an acute angle in **b**, thus the tomographic volume has been rotated to align with the T4P machinery for visualisation (upper inset box). **c** Tomographic slice through a *T. thermophilus* cell grown at 58 °C shows a wide pilus (orange arrowhead) emerging from the T4P machinery (purple arrowhead). **d** Tomographic slice of an area containing many pili from a cell preparation grown at 58 °C. Both wide (orange arrowheads) and narrow pili (teal arrowheads) are visible. OM outer membrane, PP periplasm, CM cytoplasmic membrane, CS Cytosol. Scale bars, 50 nm.

most abundant protein at 58 °C was the uncharacterised protein TT_C1836, which was present at a similar level to PilA4 at 68 °C.

To refine the identification of pilins, we performed gel-based proteomic analysis (Supplementary Fig. 2b, Supplementary Table 1). In order to increase the hyperpiliation phenotype, we further reduced the growth temperature to 55 °C. At this temperature we could identify PilA4 and TT_C1836 as the most abundant proteins in the lower molecular weight bands, likely representing the pilin monomers. In contrast, at the optimal growth temperature of 68 °C, only PilA4 was identified reliably. We questioned whether the two T4P were expressed due to differences in temperature or growth phase. To quantify the abundance of different filaments, wild-type cells were grown under different conditions and analysed in the electron microscope. Pili at both cell poles were selected for 2D classification, which enables grouping of similar structures (Supplementary Fig. 3a). At both temperatures for cells grown on plates, both wide and narrow pili were present at a similar level, whereas the ratio of the two was shifted towards the wider form for cells grown in liquid medium. At 55 °C, the total number of pili per cell increased, while the ratio between wide and narrow pili was similar to that at 68 °C (Supplementary Fig. 3b).

To analyse the role of PilA4 and TT_C1836 in pilus assembly, we investigated the number of wide and narrow pili per cell in deletion strains grown in liquid media to exponential phase (Supplementary Fig. 3c). PilA4 deficient cells (pilA4::kat) were not able to assemble any pili reliably, whereas TT_C1836 deficient cells (TT_C1836::kat) were only defective in their ability to assemble narrow pili. To ensure that the absence of pili was not due to polar effects on nearby genes, we performed RT-PCR on downstream genes (Supplementary Fig. 4a, b). The amount of transcript present was the same in wild-type, pilA4::kat and TT_C1836::kat cells. We also confirmed that the PilQ complex was still present in both mutants by Western blot analysis of total

membranes (Supplementary Fig. 4c). These findings suggest that PilA4 plays a role in producing both pilus forms, while TT_C1836 appears to be crucial for the formation of narrow pili only. However, these data do not allow us to discriminate whether the proteins have structural roles in comprising the filaments, or have a more functional role in their assembly mechanism.

**High-resolution cryoEM of both filaments.** In order to investigate the architecture and protein composition of T4P at high resolution, both filaments were subjected to analysis by single-particle cryoEM and helical reconstruction. In our micrographs, most wide pili appeared almost straight while the narrow pili were more curved (arrowhead in Supplementary Fig. 5a, b). To quantify this, we measured the curvature of filaments to determine that ~40% of narrow pili displayed a curvature higher than $2\,\mu m^{-1}$, whereas this value was only 13% for wide pili (Supplementary Fig. 5e). Based on 2D classes (Supplementary Fig. 5c, d) we determined the helical parameters for wide and narrow pili (Supplementary Fig. 6a–f, see methods section for details). The rise and twist of the wide pilus measured 9.33 Å and 92.5°, respectively, and the narrow pilus had a rise of 11.26 Å and a twist of 84.3°. Helical reconstruction[29] resulted in maps at 3.22 Å (wide) and 3.49 Å (narrow) resolution (Fig. 2a, b, Supplementary Fig. 6g, h, Supplementary Table 2). The diameter of the wide pilus is 70 Å (Fig. 2a) and is roughly cylindrical. In contrast, the narrow pilus has a zigzag-like appearance in projection, owing to a 15 Å wide groove that winds through the filament. The diameter at any position along the long axis of the filament is therefore only 45 Å (Fig. 2b). Both structures were in good agreement with the data obtained in situ by cryoET (Fig. 1). A low-resolution structure of the T4P from *T. thermophilus* was previously determined by cryoET and subtomogram averaging, with a diameter of ~3.5 nm[11]. It seems likely that this conformation represents the narrower form of the pilus.

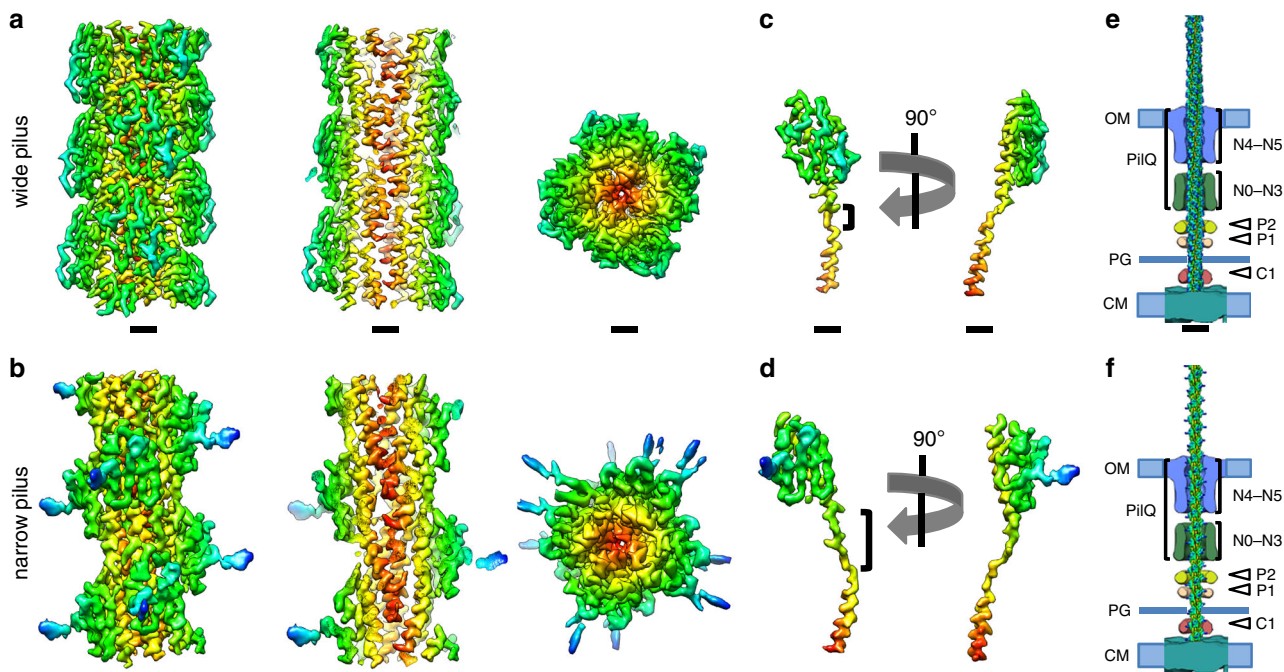

**Fig. 2 Helical reconstruction of *Thermus* pili. a, b** Side views, cross sections and top views of the density maps of wide (**a**) and narrow (**b**) pili, coloured by cylinder radius from red (centre) to blue (periphery). Scale bars, 10 Å. **c, d** Single subunits from wide (**c**) and narrow (**d**) pili segmented from the maps in **a** and **b**. The brackets show the melted stretch in the N-terminal α-helix. Scale bars, 10 Å. **e, f** Wide and narrow pili docked into the subtomogram average of the open state of the T4P machinery (EMD-3023)[11]. Domains of the secretin PilQ (blue and green; N0-N5), with unassigned protein densities P2 (yellow), P1 (orange) and C1 (red) are shown. OM outer membrane, PG peptidoglycan, CM cytoplasmic membrane. Scale bars, 10 nm.

Both maps clearly resolved individual pilin monomers. The peptide backbone could be traced throughout each subunit and large sidechains were visible (Fig. 2c, d). The centre of each pilus is formed by a bundle of long N-terminal α-helices, as has been demonstrated for other filaments[17,23–25,30]. In both maps, each α-helix is interrupted by an unfolded stretch (brackets in Fig. 2c, d), a conserved feature observed in the N-terminal domains of all available T4P structures. Interestingly, the unfolded region is significantly longer for the pilin comprising the narrow pilus, resulting in a longer N-terminal stalk. The outer regions of both filaments are formed by globular domains consisting of β-strands, a typical hallmark of the T4P C-terminal domain[31]. Whilst the C-terminal head domains of both pili are comprised of central β-sheets, the domain size and the region linking to the N-terminal α-helix appear different. These findings suggest that the two pili are not only distinct with regards to their helical parameters, but also consist of different proteins. The aperture within the central channel of PilQ is of sufficient dimension to accommodate either form (Fig. 2e, f).

**Atomic models of T4P.** The resolution and quality of both maps allowed us to unambiguously build an atomic model for each filament de novo (Fig. 3a–f). Guided by our mass spectrometry results, deletion experiments, the position of large sidechains and clear differences in the length of the polypeptide backbones, we were able to identify PilA4 as the building block for the wide pilus and the previously uncharacterised protein TT_C1836 as the subunit for the narrow filament. We now propose that TT_C1836 be named PilA5, in keeping with *Thermus* nomenclature.

The N-terminal α-helix, including the unfolded stretch, is comprised of the first 54 (PilA4) or 53 amino acids (PilA5). In both proteins the helix is disrupted by an unfolded stretch around the conserved Pro22 (Fig. 3c, d, Supplementary Fig. 7). The stretch in PilA4 is 4 amino acids long as opposed to 10 amino acids long in PilA5. The region between the N-terminal α-helix and the C-terminal β-sheet, the so-called glycosylation loop, ranges in PilA4 from amino acids 55–77, with a two-turn α-helix comprising amino acids 61–67. The C-terminal region is an antiparallel four-stranded β-sheet with the last strand facing towards the N-terminus followed by a loop that ends on the β-sheet. A disulphide bond between Cys89, which is located in the second strand, and the penultimate amino acid Cys124, likely stabilises the C-terminus (Fig. 3e). The glycosylation loop in PilA5 spans amino acids 54–71, with amino acids 62 to 65 forming a one-turn helix. The C-terminal β-sheet is composed of five strands, one more than observed in PilA4. Due to the additional β-strand in PilA5, the last strand faces away from the N-terminus of the protein. The C-terminus of PilA5 is located between the β-sheet, the glycosylation loop and the long α-helix. A disulphide bond is formed between Cys60 in the glycosylation loop and Cys88 in the third β-strand (Fig. 3f). Both pilins are highly hydrophobic at the N-terminal part of the α-helix, and more hydrophilic on the surface of the globular domain (Fig. 3g, h). The hydrophobic helices bundle and form the hydrophobic core of the assembled filament. PilA4 has no net charge but the filament displays a distinctive positively charged groove along the filament axis (Fig. 3i). In contrast, PilA5 has a total of 2 negative charges per subunit which leads to a patch of negative charge winding around the filament (Fig. 3j).

A network of cooperative interactions between pilin subunits holds the filaments together. Each subunit has 6 (wide form) or 7 (narrow form) physical interaction partners in each direction (thus 12 or 14 total interaction partners) spread in side-by-side or top-to-bottom directions (Fig. 4a, b). Most of the interactions involve a large portion of the N-terminal α-helices within the hydrophobic core as well as the head domains. In the wide (PilA4) filament, each subunit (subunit A) interacts with the N-termini that project down from the next two subunits above (B and C) and from the subunits which are six and seven subunits above (G and H). A second interaction takes place between the upper part of the α-helix in subunit A and the glycosylation loop in subunit B (Fig. 4a). In the narrow (PilA5) filaments, subunit A interacts via its α-helix with the N-termini of subunits B, C, F, G and H, while there is an additional interaction between the upper part of the α-helix in subunit A with the glycosylation loop in subunit B (Fig. 4b).

For both types of pili, the largest interaction interface is between subunits which are 3 or 4 subunits apart (subunit A with subunit D and E), which involves large parts of the N-terminal α-helices as well as the head domains. Thus each pilin subunit has a large interaction interface with 6 other subunits (B, D and E in Fig. 4a, b) and 6 or 8 smaller interaction sites (C, G, H in Fig. 4a and C, G, H, F in Fig. 4b). Most interactions involve the hydrophobic sidechains in the centre of the filament and appear to be nonspecific, likely allowing sliding movements between the subunits when the filaments are bent or stretched. This is in accordance with the observation that pili can stretch up to threefold upon force[32]. PilA4 contains two intermolecular salt bridges—between Asp53 (subunit A) and Arg30 (subunit D), and between Glu48 (subunit A) and Arg28 (subunit E) (Fig. 4c). In contrast, PilA5 contains a single intermolecular salt bridge—between Glu68 (subunit A) and Arg23 (subunit D) (Fig. 4d). Intramolecular salt bridges are also observed in PilA4 between Asp42 and Lys45, and between Lys107 and the C-terminus (Pro125), and in PilA5 between Asp37 and Arg104, and Asp81 and Arg99 (Fig. 4c, d). A conserved Glu5 in both filaments is likely required to neutralise the positive charge of the N-terminus within the hydrophobic core[4,33,34]. In other structures of either pilins or T4P, a salt bridge has been modelled between Glu5 and the N-terminus of the neighbouring subunit[17,23,24], which is probably methylated[16,35]. The resolution of our maps is not sufficient to show either salt bridges or methylation at the N-terminus. Therefore, we measured the distance between the N-termini and all potentially negatively charged residues in both models. In both filaments, Glu5 is the nearest negatively charged residue to the N-terminus of the same subunit (Supplementary Table 3), hence most likely forms an additional intramolecular salt bridge with Phe1 (Fig. 4c, d). An intramolecular salt bridge between Glu5 and the N-terminus has also been observed previously[36–38]. It has been shown that N-terminal methylation and an intramolecular salt bridge can reduce the interaction of pilins with the plasma membrane, which might be beneficial during pilus assembly[38].

**Post-translational modification.** Densities were observed in both EM maps that protrude into the solvent and cannot be attributed to the polypeptide backbone and were too large to account for amino acid sidechains (Fig. 5a, b). These densities colocalised with serine residues and are similar in appearance to previously published densities attributed to glycans[30,39,40]. Moreover, the bands of both pilins could be detected in SDS gels by staining periodate-oxidized carbohydrate groups, which is consistent with the previous finding that the major pilin PilA4 of *T. thermophilus* is glycosylated[41] (Fig. 5c). In PilA4, we found extra densities at 3 serine residues in the glycosylation loop (Ser59, Ser66 and Ser71), while only one serine appeared to be modified in PilA5 (Ser73). Glycosylation has also been observed in similar locations in the X-ray structure of the *N. gonorrhoeae* type IV pilin head domain[16,17,42]. Interestingly, the density is much more pronounced in PilA5 than in PilA4. This may be due to a less flexible

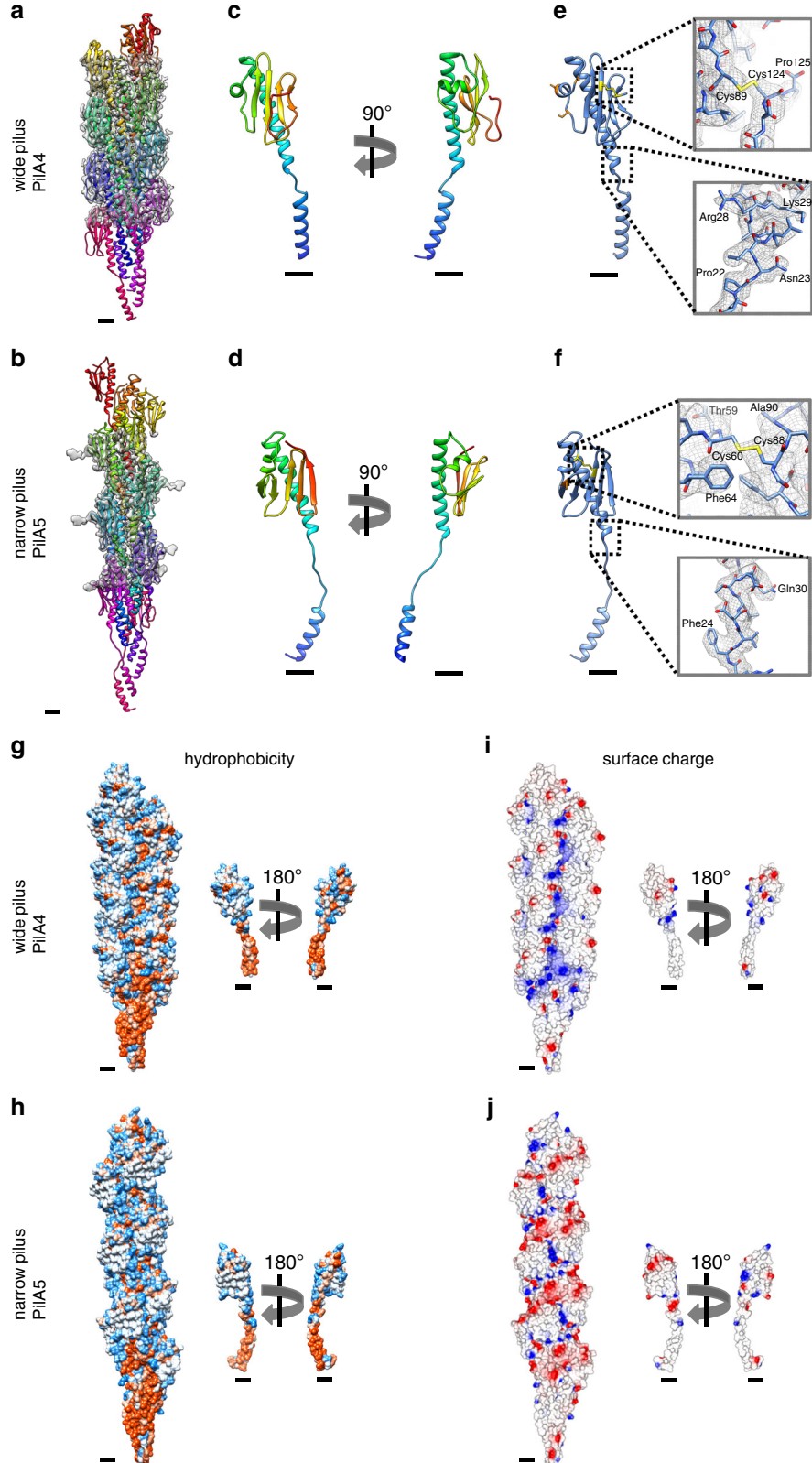

**Fig. 3 Atomic models of wide and narrow pili. a**, **b** Models of short sections of filaments (15 subunits each) with the corresponding EM density maps.
**a** wide pilus comprised of PilA4; **b** narrow pilus comprised of PilA5. Scale bars, 10 Å. **c**, **d** Ribbon representation of a single PilA4 (**c**) and PilA5 (**d**) subunit.
N-terminus blue, C-terminus red. Scale bars, 10 Å. **e**, **f** Ribbon representation of a single PilA4 (**e**) and PilA5 (**f**) subunit with selected sidechains shown.
Yellow, disulphide bond; orange, serines with post-translational modification. Insets show details of the atomic model within the EM density map. Top
insets, disulphide bond; bottom insets, side chain densities. Scale bars, 10 Å. **g**, **h** Hydrophobicity patterns of PilA4 (**g**) and PilA5 (**h**) filaments (left) and
single subunits (right). Hydrophobic (red) and hydrophilic (blue) residues are shown. Scale bars, 10 Å. **i**, **j** Electrostatic surface charge of PilA4 (**i**) and PilA5
(**j**) filaments (left) and single subunits (right). Negative charges (red) and positive charges (blue) are shown. Scale bars, 10 Å.

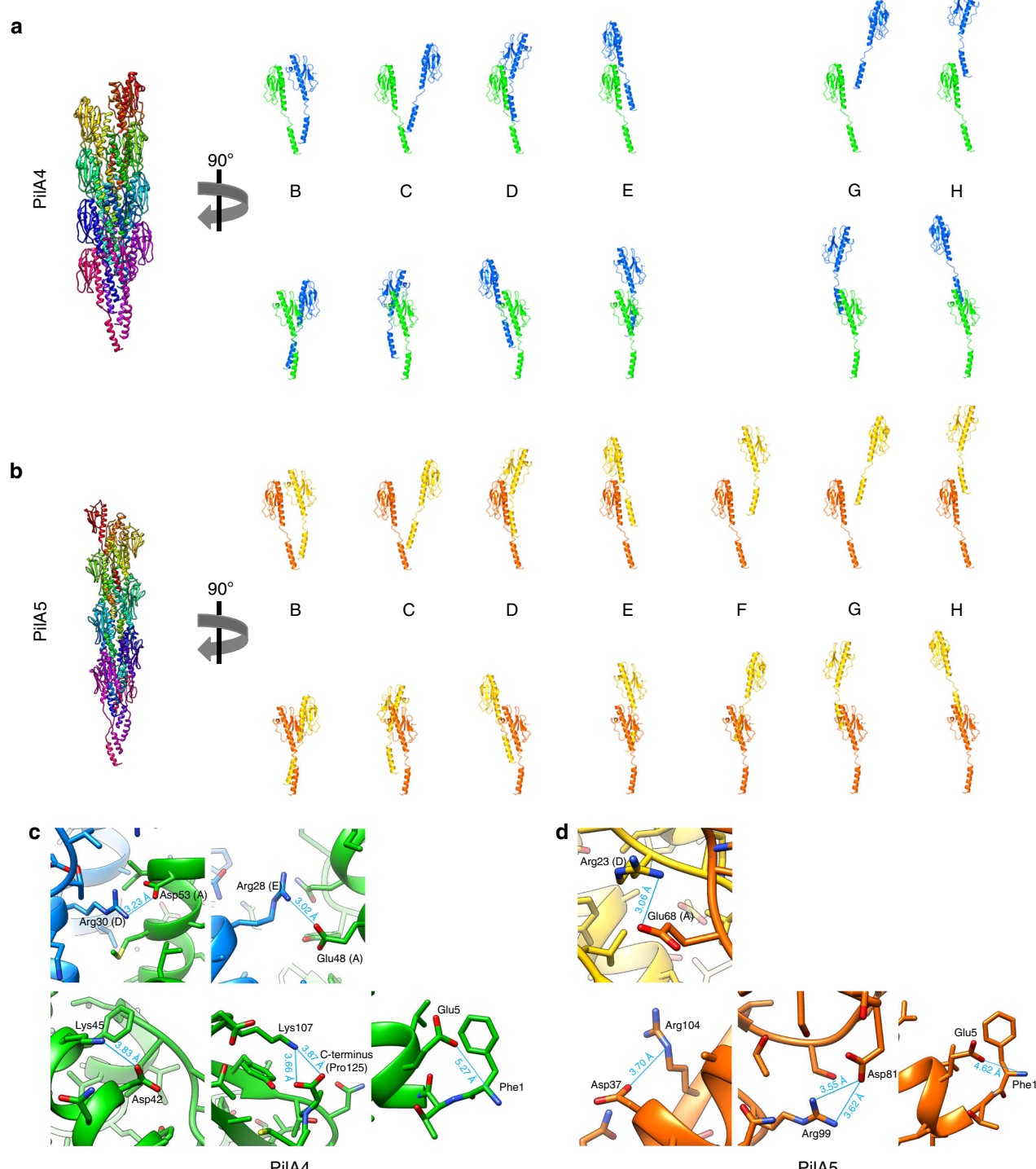

**Fig. 4 Intra and intermolecular interactions. a** Ribbon representation of a 15-mer wide filament (left) and pairs of interacting subunits. Subunit A (green) is shown with other single subunits within the filament (blue) from B (+1) to H (+7). Subunit F (+5) is not shown because it does not interact with subunit A. **b** Ribbon representation of a 15-mer narrow filament (left) and pairs of interacting subunits. Subunit A (orange) is shown with other single subunits within the filament (yellow) from B (+1) to H (+7). **c, d** Close-ups of intermolecular (top row, letters in brackets correspond to the subunits shown in **a**, **b**) and intramolecular (bottom row) salt bridges within PilA4 wide (**c**) and PilA5 narrow (**d**) filaments. Same colour code as in **a**, **b**.

glycan moiety in PilA5, allowing for improved resolution, or by a different composition of the sugar residues. Lectin-based detection of glycans suggests the presence of N-acetylgalactosamine and mannose and/or glucose on both filaments (Fig. 5d). In order to confirm the presence of glycans we performed detailed glycoprofiling experiments. The O-glycans from wild-type samples

containing both wide and narrow pili were released using anhydrous hydrazine, labelled with procainamide, separated and analysed by HILIC-UHPLC, and glycan composition was subsequently analysed by coupled ESI-MS and MS/MS. These data showed that filaments can be modified by a number of O-linked glycans containing the following monosaccharide residues:

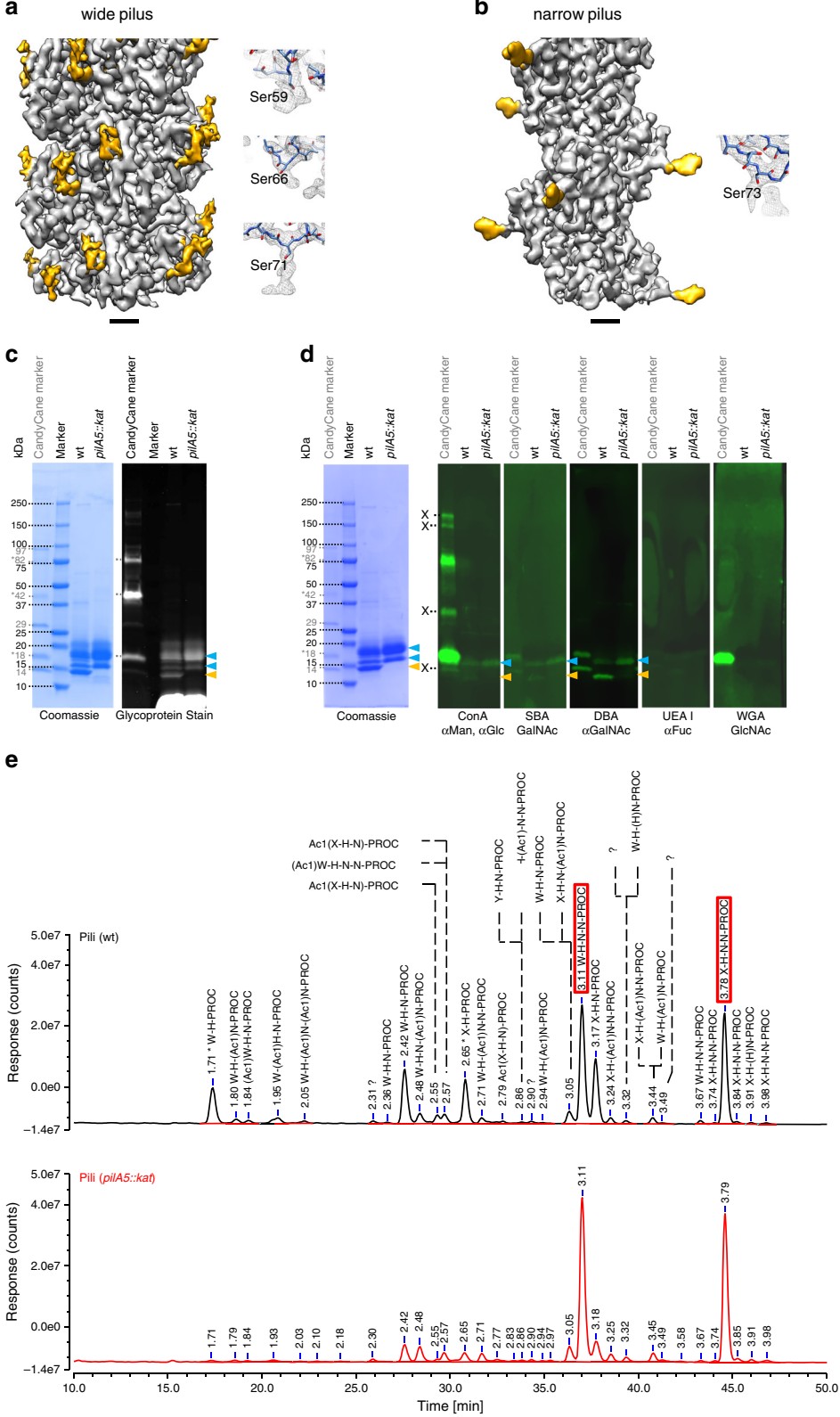

hexose (H), N-acetylhexosamine (N), Pseudaminic acid derivative 5Am7Ac (W, 7-Acetamido-5-acetimidoyl-3,5,7,9-tetradeoxy-L-glycero-L-manno-nonulosonic acid), as well as two unknown and previously unreported monosaccharides (Fig. 5e, Supplementary Table 4). Different monosaccharide residues were also shown to be acetylated. The procainamide labelled carbohydrates present in

the highest amount, and therefore that most likely account for the electron density in our maps, are W-H-N-N and X-H-N-N (Supplementary Fig. 8). X represents an unknown carbohydrate unit of 346 Da and does not correspond to any previously reported bacterial structures. Comparison of these data and the corresponding GU (glucose unit) values[43,44] to the data generated

**Fig. 5 Both wide and narrow pili are glycosylated. a, b** Surface representation of the EM maps of wide (**a**) and narrow (**b**) pili showing the protein model (grey) and densities that protrude into the solvent (yellow) that could not be attributed to the polypeptide backbone or an amino acid side chain. Insets show close-ups of large unassigned densities near serine residues. Scale bars, 10 Å. **c** Coomassie and glycoprotein stain of isolated pili. Pili isolated from cells grown at 55 °C were purified and separated by SDS-PAGE. Total protein was Coomassie-stained (left panel) and glycoproteins were stained using Pro-Q Emerald 300 Glycoprotein Stain (right panel). Bands for PilA4 (blue) and PilA5 (orange) are marked. Bands in the CandyCane marker are shown in grey and glycosylated bands are highlighted with an asterisk. **d** Lectin analysis of isolated pili. Pili isolated from cells grown at 55 °C were purified and separated by SDS-PAGE. For comparison a Coomassie-stained gel is shown (left). Proteins were transferred to PVDF membranes and labelled with biotinylated lectins and IRDye 800CW Streptavidin (right). Bands for PilA4 (blue) and PilA5 (orange) are shown. Bands in the CandyCane marker are marked in grey and glycosylated bands are highlighted with an asterisk. Possible cross-reactivity of the lectins with the marker is highlighted with [X]. The primary sugar specificity is indicated with the name of the lectins. ConA, Concanavalin A (Man, mannose; Glc, D-glucose); SBA, soybean agglutinin (GalNAc, N-acetylgalactosamine); DBA, *Dolichos biflorus* agglutinin (GalNAc, N-acetylgalactosamine); UEA I, *Ulex europaeus* agglutinin I (Fuc, L-fucose); WGA, wheat germ agglutinin (GlcNAc, N-acetylglucosamine). **e** Glycoprofiling of pili. Comparison of UHPLC profiles generated from a wild-type (wt) sample (top, containing both wide and narrow pili) and a *pilA5::kat* sample (bottom, containing wide pili only). Glycan structures were identified in the wt sample by ESI-MS/MS fragmentation. The two most abundant glycans W-H-N-N-PROC and X-H-N-N-PROC are highlighted with a red box. H, hexose; N, N-acetylhexosamine; W, Pseudaminic acid derivative 5Am7Ac (7-Acetamido-5-acetimidoyl-3,5,7,9-tetradeoxy-L-glycero-L-manno-nonulosonic acid); X and Y, unknown and previously unreported monosaccharides with a molecular mass of 346 Da (X) and 330 Da (Y); PROC, procainamide; Ac, acetylation; question marks label glycans with unassigned structure due to insufficient MS/MS data. Source data are shown in Supplementary Fig. 10 (panels **c**, **d**) and in a Source Data file (panels **c**–**e**).

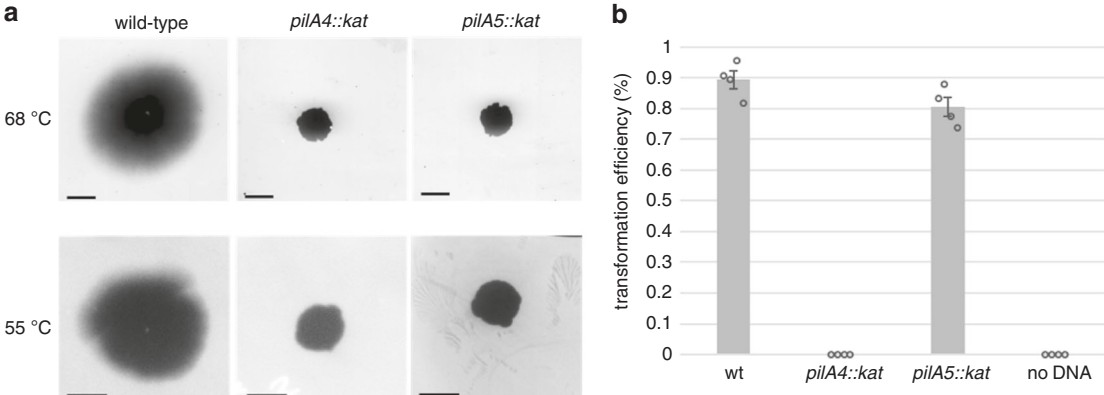

**Fig. 6 Functional characterisation of *T. thermophilus* mutants. a** Twitching motility of *T. thermophilus* HB27 strains. Only wild-type cells show twitching. Scale bars = 0.5 cm. **b** Natural transformation efficiency of *T. thermophilus* HB27 strains. Data are presented as mean values ± SEM. Source data are shown in Supplementary Fig. 10 (panel **a**) and in a Source Data file (both panels).

from the sample containing wide pili only, suggests again that the structures present in the highest amounts were W-H-N-N and X-H-N-N, eluting at GU 3.11 and GU 3.78 respectively (Fig. 5e). Based on our lectin-binding experiments (Fig. 5d), it is likely that the N-acetylhexosamine is GalNAc, and the hexose is glucose or mannose. Glycosylation may prevent pilus bundling, enhance temperature stability via additional hydrogen bonds, increase adhesive properties (either to surfaces or to molecules such as DNA) or act as recognition tags for cell-cell communication[45].

**The functional importance of two types of pili.** A key outstanding question pertains to the functional relevance of the two types of T4P. To investigate this question, we performed various functional analyses on PilA4 and PilA5 deletion strains. We assessed cell lines without pili (*pilA4::kat*), with wide pili only (*pilA5::kat*), or a mixed population of wide and narrow pili (wild-type).

We analysed cellular motility by twitching assays at 68 °C and 55 °C (Fig. 6a). Wild-type cells formed characteristic twitching zones of ~2 cm and ~1.2 cm in diameter, respectively. The mutants *pilA4::kat* and *pilA5::kat* did not exhibit any twitching motility. Since the immotile *pilA5::kat* cells could still produce wide pili comprised of PilA4, we deduce that PilA5 is required to promote cell movement. Cells lacking both types of pili in the *pilA4::kat* mutant were completely defective in natural

transformation, in agreement with our former finding[26]. Transformation efficiency was only partially reduced in the *pilA5::kat* mutant (~10%), which expresses wide PilA4 pili (Fig. 6b). This corresponds to our previous finding that a *pilA5::kat* mutant is still transformable[26] and demonstrates that the narrow pili are dispensable for DNA uptake.

In summary, we conclude that PilA4 has three known roles: it promotes pilus formation for both wide and narrow filaments, it comprises the main structure of the wide T4P, and it plays a role in natural transformation. We determine two functions for PilA5: it forms the main structure of the narrow T4P and is a requirement for cell motility.

## Discussion

We have determined the first cryoEM structures of T4P that have allowed atomic models to be built de novo. Moreover, we have discovered two distinct T4P filaments, which are composed of different proteins. Both filaments are decorated with O-linked glycan chains, and we detect an unknown carbohydrate structure not seen before in bacteria. Our data provide compelling evidence that PilA5 is essential for twitching motility and confirm the previous finding that PilA4 is involved in natural transformation[26]. In addition, we find that PilA4 is essential for assembly of both the wide and narrow pili. PilA4 may therefore play a crucial regulatory role, or be required for stability of the entire

machinery. In many bacteria, minor pilins are thought to prime pilus assembly by reducing the energy barrier to the extraction of pilins from the membrane[15,46]. In *Thermus*, PilA4 may perform this role.

The unique functionality of PilA4 and PilA5 is hardcoded in their distinct structural features. Both filaments follow the conserved T4P blueprint, encompassing a central bundle of hydrophobic N-terminal α-helices and a hydrophilic C-terminal β-strand globular domain[45]. Structural variations in PilA4 and PilA5 determine distinct inter-subunit interactions, helical parameters, mechanical and adhesive properties. Supported by our curvature analysis, narrow pili comprised of PilA5 appear to be more flexible than those comprised of PilA4. In line with their predicted role in twitching, narrow pili that can bend and flex would enable the filaments to curve from the surface of cells to interact with surfaces, negotiate obstacles and increase the exploratory range of the cell. Their overall net negative surface charge would enhance the adhesive properties of the filament and facilitate surface adhesion.

T4P are well known to play a role in DNA binding and uptake in different species[47,48]. Craig et al. suggest that the positively charged groove of gonococcal T4P is wide enough to bind the negatively charged backbone of dsDNA[17]. The surface of our wide PilA4 filaments show a striking line of positively charged residues along the long axis, into which a double stranded DNA molecule can be fitted (Supplementary Fig. 9). However, recent studies have shown that the tip of the filament is important for DNA binding in *Vibrio cholerae*[49], and it has been suggested that the *Thermus* filament cap may be comprised of the minor pilin PilA2 and ComZ, a DNA-binding protein[50]. In addition, functional experiments in *Thermus* have shown that pili are not absolutely required for natural transformation. For example, a mutant defective in the PilF assembly ATPase was impaired in piliation but was surprisingly hypertransformable, and a mutant carrying a deletion in a domain of the secretin PilQ was impaired in piliation but exhibited wild-type transformation frequencies[10,51]. Taken together, we suggest that wide pili comprised of PilA4 may capture DNA rather like a fishing net, thus improving the efficiency of DNA uptake by increasing the local concentration of DNA near the outer membrane.

In evolutionary terms, bacteria appear to have reused the T4P blueprint to develop a system that can assemble two different filaments with unique properties. This could enable tasks to be performed more effectively and at reduced energy cost to the cell. Interestingly, filaments of different diameter have also been observed on the surface of *Synechocystis*[52]. It will now be important to discover if this principle occurs in other bacterial species, and excitingly, will open avenues to the development of vaccines or therapeutics targeting a particular T4P mechanism.

## Methods

**Cultivation of organisms**. *T. thermophilus* HB27 was grown in TM$^+$ medium (8 g/l tryptone, 4 g/l yeast extract, 3 g/l NaCl, 0.6 mM MgCl$_2$, 0.17 mM CaCl$_2$)[53] at 55 °C, 58 °C or 68 °C. Antibiotics were added when appropriate (kanamycin, 80 mg/ml; streptomycin, 100 mg/ml in solid medium (containing 2% agar [w/vol]) or kanamycin, 60 mg/ml; streptomycin 100 mg/ml in liquid medium). Disruption of *TT_C1836* and *pilA4* was performed by insertion of a kanamycin marker[27]. *TT_C1836::kat = pilA5::kat*, which was formerly designated *Tt17*.

**Purification of pili**. *T. thermophilus* HB27 cells were grown for two days on TM$^+$ medium at 68 °C or three days at 55 °C. Cells were scraped off and resuspended by pipetting and shaking in ethanolamine buffer (0.15 M ethanolamine, pH 10.5). Cells were sedimented by centrifugation (16,200 × *g*, 1 h, 4 °C). The supernatant was gently mixed with saturated ammonium sulfate solution [10/1 (v/v)] and incubated on ice for 12 h to precipitate pili. Pili were sedimented by centrifugation for 10 min (16,200 × *g*, 4 °C). The resulting pellet was washed twice with TBS buffer (50 mM Tris/HCl, 150 mM NaCl, pH 7.5). The pellet was resuspended by

incubation in distilled water for 4 h. 10x buffer (500 mM Tris/HCl, 500 mM NaCl, 10 mM CaCl$_2$, 10 mM MgCl$_2$, pH 7.5) was added prior to structural analyses.

**Biochemical analysis of purified pili**. Purified pili and molecular weight markers (Precision Plus Protein™ All Blue Prestained Protein Standards, Biorad, Hercules, USA; CandyCane glycoprotein molecular weight standards, Thermo Fisher, Waltham, USA) were separated by SDS-PAGE (Mini Protean TGX 4–20%, Biorad). Protein was stained using InstantBlue™ Protein Stain (Expedeon Ltd, Cambridge, UK) or Pro-Q™ Emerald 300 Glycoprotein Gel and Blot Stain Kit (Thermo Fisher). For the lectin analysis, after SDS-PAGE proteins were transferred to PVDF membranes (Trans-Blot Turbo System, Biorad). Blots were blocked and incubated with biotinylated lectins (10 μg/ml; Vector Laboratories, Burlingame, USA) according to the manufacturer's instructions, followed by incubation with IRDye 800CW Streptavidin (LI-COR, Lincoln, USA) and imaging using an Odyssey Fc (LI-COR).

**RNA-preparation and RT-PCR**. *T. thermophilus* cells were grown in 45 ml TM$^+$ -medium at 68 °C until mid-exponential phase (OD 0.6). Cells were harvested by centrifugation (4100 × *g*, 15 min, 4 °C). Cells were resuspended in 700 μl RLT buffer (Qiagen, Hilden, Germany) containing 1% β-mecaptoethanol and lysed by bead-beating with 0.1 mm diameter zirconia beads (Sigma-Aldrich, St Louis, USA) for 2 × 3 min in RLT buffer containing 1% β-mecaptoethanol for 2 × 3 min. RNA was prepared using the "RNeasy® Mini Kit" (Qiagen) according to the manufacturer's instructions. Four microgram of the resulting RNA was used for DNase digestion using the RQ1 DNase (Promega, Madison, USA) according to the manufacturer's instructions. One microgram of DNase treated RNA was used for reverse transcription using the M-MLV reverse transcriptase (Promega) with random primers (Promega) according to manufacturer's specifications. Hundred nanogram of cDNA was used as template for PCR amplification. For amplification of *TT_C0859*, located directly downstream of *pilA4*, TTC_0859_for (CAGGTGC GGCCCCTGACCTTGG) and TTC_0859_rev (TTACAGGCCGTGGTACGGCCT CCG) primers were used. For amplification of *TT_C0860* located 2432 bps downstream of the *pilA4* translational stop, TTC_0860_for (TCTAACTACCCGGACC TCATGGTGGTTTGC) and TTC_0860_rev (ATTATACAGGCCGTGGTACGCC TCCG) primers were used. To amplify *TT_C1837*, located 23 bps downstream of *pilA5*, the primers TTC_1837_for (TGCGGCTTAAACCTCTCCC) and TTC_ 1837_rev (TCCTCCCGCGTGACCAG) were used.

**Preparation of total membranes and SDS-PAGE**. *T. thermophilus* cells were grown in 45 ml TM$^+$-medium at 68 °C or 55 °C overnight and harvested by centrifugation (4100 × *g*, 15 min, 4 °C). The cell pellets were resuspended in 30 ml of membrane buffer (40 mM Tris/HCl, 200 mM NaCl, 5 mM CaCl$_2$, 5 mM MgCl$_2$, pH 8.5) and cells were lysed by sonication (45 Hz, 2 × 3 min). Cell debris was removed by centrifugation (11,000 × *g*, 15 min, 4 °C). Membranes were separated by ultracentrifugation (260,000 × *g*, 45 min, 4 °C) and resuspended in 200 μl of membrane buffer. Twenty microgram protein from total membrane fractions were separated on a 3–12% gradient SDS-PAGE gel and stained with InstantBlue™ Protein Stain (Expedeon Ltd, Cambridge, UK). To verify the production of PilQ complex, proteins were transferred from the SDS gel onto a nitrocellulose membrane and Western blot analysis was performed with polyclonal PilQ antibodies (1:6000)[41].

**CryoET sample preparation and imaging**. Cubes of agar with growing *T. thermophilus* HB27 cells were cut out, placed into EDTA buffer (20 mM Tris/HCl, 100 mM EDTA, pH 7.4) and gently agitated for 1 h at room temperature. Samples were mixed 1:1 with 10 nm protein A-gold (Aurion, Wageningen, The Netherlands) as fiducial markers and glow-discharged R2/2 Cu 300 mesh holey carbon-coated support grids (Quantifoil, Jena, Germany) were dipped into the solution. For analysis of isolated pili by cryoET, preparations were mixed 1:1 with 10 nm protein A-gold fiducial markers and solutions were gently pipetted onto the grids. Grids were plunge-frozen in liquid ethane using a home-built device (blot time: 4–6 s, blotting paper: Whatman 41) in a humidified atmosphere.

Tomograms were typically collected from +60° to −60° at tilt steps of 2° and 5–7 μm underfocus (whole cells), or at 3 μm underfocus (isolated pili), using either a Tecnai Polara, Titan Krios (Thermo Fisher) or JEM-3200FSC (JEOL, Tokyo, Japan) microscope, all equipped with field emission guns operating at 300 keV. All instruments were fitted with energy filters and K2 Summit direct electron detector cameras (Gatan, Pleasanton, USA). Dose-fractionated data (3–5 frames per projection image) were collected using Digital Micrograph (Gatan). Magnifications varied depending on microscope; pixel sizes were within the range 3.8–4.2 Å. The total dose per tomogram was <~140 e$^-$/Å$^2$. Tomograms were aligned using gold fiducial markers and volumes reconstructed by weighted back-projection using the IMOD software (Boulder Laboratory, Boulder, USA)[54]. Contrast was enhanced by non-linear anisotropic diffusion (NAD) filtering in IMOD[55].

**Negative stain electron microscopy**. Two microliters of purified pili were pipetted onto glow-discharged carbon-coated Cu 400 mesh support grids (Sigma-Aldrich) for 2 min. Grids were blotted with Whatman No 41 filter paper and stained with 5% ammonium molybdate for 60 s. Images were recorded with a

Tecnai Spirit microscope (Thermo Fisher) operated at 120 keV and a OneView CMOS camera (Gatan). Images were analysed for pilus quality, size, sample density and homogeneity using EMAN2[56]. For whole cell samples of *T. thermophilus*, either liquid culture was used directly or some cells were carefully scraped off the plates and resuspended in TBS. If required, cells were diluted in TBS. Negative staining was performed as described above. For the quantitative analysis of number and type of pili per cell, filaments from each cell pole were counted and helices of equal length were selected using e2helixboxer (EMAN2) and subsequently classified in 2D using RELION[57]. The percentage of wide, narrow and unassigned pili was calculated based on the number of particles in each class.

**CryoEM sample preparation and imaging of pili**. Three microliters of isolated pilus suspension were pipetted onto a glow-discharged R2/2 Cu 300 mesh holey carbon-coated grids (Quantifoil). Grids were plunge-frozen in liquid ethane after blotting using a Vitrobot Mark IV (Thermo Fisher) (wait time: 30 s, blot time: 3.5 s, drain time: 0 s, blot force: −1, blotting paper: Whatman 597, chamber temperature: 16 °C, humidity: 95%) and stored in liquid nitrogen. Cryo images were collected with a Titan Krios microscope (Thermo Fisher) at the UK Electron Bio-Imaging Centre (eBiC), equipped with a field emission gun operating at 300 keV. The microscope was fitted with K2 Summit direct electron detector and Quantum energy filter (both Gatan). Dose-fractionated data were collected at 1.5–4 µm defocus using EPU (Thermo Fisher). 3,138 micrographs containing both forms of pili were collected as 40-frame movies, corresponding to 8 seconds at a frame rate of 1 frame for every 0.2 seconds. The total dose was 48 electrons/Å² at a magnification of 130,000x, corresponding to a pixel size of 1.048 Å. For further details see Supplementary Table 2.

**Curvature analysis of pili**. Wide and narrow pili were traced in cryoEM micrographs using ImageJ[58] and the plugin Kappa. The curvature data of three independent subsets of images with more than 60 wide and narrow filaments each were further processed to create a histogram.

**Image processing and helical reconstruction**. Drift correction was performed using UNBLUR[59]. Straight sections of wide and narrow pili were boxed separately from the drift corrected images using the helixboxer function of EMAN2, such that the filaments were centred in each rectangular box. Helical reconstruction was performed using the boxed filaments and SPRING as follows[60]. Contrast transfer function (CTF) correction was performed using CTFFIND[61]. In order to determine the helical parameters of both pili, a subset of boxed filaments were cut into small segments of varying box size and overlap, which were classed in 2D (Supplementary Fig. 5c, d). For wide pili, close examination of the segments indicated a filament diameter of 75 Å. The calculated power spectrum from the total segments indicated clear layer lines that could be indexed (Supplementary Fig. 6a). A meridional reflection at approximately 9 Å and a layer line of order 1 at ~36 Å indicated that there are ~4 subunits per turn. The ninth layer was found to be of order 1, suggesting that the helix repeats exactly after nine turns, with a non-integer number of subunits per turn. The SEGMENTCLASSRECONSTRUCT module in SPRING was used to determine the accurate helical symmetry (Supplementary Fig. 6b). The suggested output was determined to be either 4.10 or 3.89 subunits in a helical pitch of 36.3 Å. For narrow pili, the helical pitch could be determined directly as 48.1 Å from the 2D classes. A meridional reflection and thus a helical rise at 11.3 Å could be identified from the power spectrum (Supplementary Fig. 6d). These parameters allow calculation of a helical rotation of 84.6° and 4.26 subunits per turn. The SEGMENTCLASSRECONSTRUCT module in SPRING was again used to determine the accurate helical symmetry (Supplementary Fig. 6e). The suggested output was determined to be either 4.11, 4.14, 4.27 or 4.30 subunits with a helical pitch of 48.1 Å. 3D reconstruction was performed using the above parameters by iterative projection matching and back projection as implemented in the SEGMENTREFINE3D of SPRING, starting from a solid cylinder of 80 Å as a reference. Examination of the Fourier transforms simulated from the reconstructed volume to that experimentally calculated indicated that 3.89 subunits in a pitch of 36.3 Å (accounting for a helical rise of 9.33 Å and a helical rotation of 92.5 degrees) is correct for the wide filaments, and 4.27 subunits in a pitch of 48.1 Å (accounting for a helical rise of 11.26 Å and a helical rotation of 84.3 degrees) is correct for the narrow filaments (Supplementary Fig. 6c, f). For the final maps 300 Å segments with a step size of three times the helical rise from 400 images were extracted. Doubling the number of used images did not further increase the final resolution. The calculated final maps were determined at 3.22 Å resolution from 65,656 segments (196,968 asymmetric units) for wide filaments, and 3.49 Å from 51,301 segments (153,903 asymmetric units) for narrow filaments (Supplementary Fig. 6g, h, Supplementary Table 2) using Fourier shell correlation (0.143 cut-off). For the final maps a B-factor of −60 Å² was applied. Figures were drawn in Chimera, Coot and CCP4mg[62–64].

**Model building**. Atomic models for both forms of pili were built manually de novo in Coot. We assumed that one of the two pili consists of the major pilin PilA4, which is 125 amino acids in length. The backbone as well as all large side chain densities of PilA4 match the density map of the wider form of the pilus. While tracing the backbone into the density maps it became apparent that the subunits

forming the narrower pili are ~10% smaller than the subunits of the wider pili. Following the results of mass spectrometry and deletion experiments we modelled the second most abundant protein, TT_C1836 (111 amino acids) into the density map of thin pili. The backbone and visible sidechains fit perfectly into the density map. First, the structure of the monomers were iteratively refined by Refmac5[65] followed by manual rebuilding in Coot and ISOLDE[66]. For later refinement iterations a filament of 16 subunits was created by applying the helical symmetry to the monomer followed by refinement of the filament by Refmac5. Since only the central subunits of the filament were completely within the cryoEM map, in the next step only the central subunit (chain A) was rebuilt in Coot and a new 16-mer filament based on the refined chain A was created by applying helical symmetry. The final models contain 16 identical subunits with all amino acid residues of the mature protein. The double stranded DNA in Supplementary Fig. 9 (based on PDB: 1bna) was modelled around the wide pilus using Chimera and Coot.

**Sequence alignment**. Sequence alignment was performed using the PRALINE server[67] with the default settings (weight matrix: BLOSSUM62, gap opening penalty: 12, gap extension penalty: 1).

**Protein identification by mass spectrometry (MS)**. Purified pilus preparations were processed using a modified filter-aided sample preparation (FASP) workflow[68] as described previously[69]. In brief, reduced and alkylated protein extracts were digested sequentially with Lys-C and trypsin on Microcon-10 filters (Merck, Burlington, Massachusetts, USA, # MRCPRT010 Ultracel YM-10). Digested samples were desalted using ZipTips according to the manufacturer's instructions, dried in a Speed-Vac and stored at −20 °C until liquid chromatography with tandem mass spectrometry (LC/MS-MS) analysis. Dried peptides were dissolved in 5% acetonitrile with 0.1% formic acid, and subsequently loaded using a nano-HPLC (Dionex U3000 RSLCnano) on reverse-phase columns (trapping column: particle size 3 µm, C18, L = 20 mm; analytical column: particle size < 2 µm, C18, L = 50 cm; PepMap, Dionex/Thermo Fisher). Peptides were eluted in gradients of water (buffer A: water with 5% v/v acetonitrile and 0.1% formic acid) and acetonitrile (buffer B: 20% v/v water and 80% v/v acetonitrile and 0.1% formic acid). All LC-MS grade solvents were purchased from Fluka (Sigma-Aldrich). Gradients were ramped from 4 to 48% B in 120 min at flow rates of 300 nl/min. Peptides eluting from the column were ionised online using a Thermo nanoFlex electrospray ionization (ESI)-source and analysed in a Thermo "Q Exactive Plus" mass spectrometer. Mass spectra were acquired over the mass range 350–1400 *m/z* (Q Exactive Plus) and sequence information was acquired by computer-controlled, data-dependent automated switching to MS/MS mode using collision energies based on mass and charge state of the candidate ions.

Raw MS data were processed and analysed with MaxQuant[70]. In brief, spectra were matched to the full *Thermus thermophilus* HB27 NCBI nr database (reviewed and non-reviewed, downloaded on the 13/05/2016) and a contaminant and decoy database. Precursor mass tolerance was set to 4.5 ppm, fragment ion tolerance to 20 ppm, with fixed modification of Cys residues (carboxyamidomethylation + 57.021) and variable modifications of Met residues (Ox + 15.995), Lys residues (Acetyl + 42.011), Asn and Gln residues (Deamidation + 0.984) and of N-termini (carbamylation + 43.006). Peptide identifications were calculated with a false discover rate (FDR) = 0.01, and proteins with one peptide per protein included for subsequent analyses. Peptide intensities (label-free quantitation) were analysed using MaxQuant and Perseus[70]. Differential abundance of proteins (detected in at least 3 of 4 replicates in each condition) was analysed using a two-sided t-test with an FDR of 0.01 and s0 = 0.05.

For gel-based MS, purified pili were separated by SDS-PAGE (Mini Protean TGX 4–15%, Biorad) and proteins were stained using Bio-Safe Coomassie Stain (Biorad). Gel bands were subjected to in-gel tryptic digestion using a DigestPro automated digestion unit (Intavis Ltd, Köln, Germany). The resulting peptides were fractionated using an Ultimate 3000 nano-LC system in line with an LTQ-Orbitrap Velos mass spectrometer (Thermo Fisher). In brief, peptides in 1% (vol/vol) formic acid were injected onto an Acclaim PepMap C18 nano-trap column (Thermo Fisher). After washing with 0.5% (vol/vol) acetonitrile 0.1% (vol/vol) formic acid peptides were resolved on a 250 mm × 75 µm Acclaim PepMap C18 reverse-phase analytical column (Thermo Fisher) over a 60 min organic gradient, (1–50% solvent B over 45 min, 50–90% B over 0.5 min, held at 90% B for 4.5 min and then reduced to 1% B over 0.5 min) with a flow rate of 300 nl min⁻¹. Solvent A was 0.1% formic acid and Solvent B was aqueous 80% acetonitrile in 0.1% formic acid. Peptides were ionized by nano-electrospray ionization at 2.1 kV using a stainless steel emitter with an internal diameter of 30 µm (Thermo Fisher) and a capillary temperature of 250 °C. Tandem mass spectra were acquired using an LTQ- Orbitrap Velos mass spectrometer controlled by Xcalibur 2.1 software (Thermo Fisher) and operated in data-dependent acquisition mode. The Orbitrap was set to analyze the survey scans at 60,000 resolution (at *m/z* 400) in the mass range m/z 300 to 2000 and the top twenty multiply charged ions in each duty cycle selected for MS/MS in the LTQ linear ion trap. Charge state filtering, where unassigned precursor ions were not selected for fragmentation, and dynamic exclusion (repeat count, 1; repeat duration, 30 s; exclusion list size, 500) were used. Fragmentation conditions in the LTQ were as follows: normalized collision energy, 40%; activation q, 0.25; activation time 10 ms; and minimum ion selection intensity, 500 counts.

The raw data files were processed and quantified using Proteome Discoverer 1.4 (Thermo Fisher) and searched against the UniProt *Thermus thermophilus* strain HB27 [262724] database (downloaded August 2018: 2201 sequences) using the SEQUEST algorithm. Peptide precursor mass tolerance was set at 10 ppm, and MS/MS tolerance was set at 0.8 Da. Search criteria included carbamidomethylation of cysteine (+57.0214) as a fixed modification and oxidation of methionine (+15.9949) as a variable modification. Searches were performed with full tryptic digestion and a maximum of 1 missed cleavage was allowed. The reverse database search option was enabled and all peptide data was filtered to satisfy an FDR of 1%.

**Glycoprofiling by LC-ESI-MS and MS/MS analysis**. All reagents and kits for O-glycan release, procainamide labelling and clean-up were from Ludger Ltd (Oxford, UK). Samples of wide and narrow pili (wild-type), wide pili only (*pilA5::kat*), and fetuin glycoprotein (used as a positive control) were buffer exchanged into 0.1% trifluoroacetic acid (TFA) prior to hydrazinolysis as described previously[71]. Briefly, samples were transferred to glass vials and dried down for 16 h by vacuum centrifugation prior to the addition of anhydrous hydrazine at 60 °C for 6 h[72,73]. Excess hydrazine was removed by centrifugal evaporation. The samples were placed on ice for 20 min (0 °C) and were re-*N*-acetylated by addition of 0.1 M sodium bicarbonate solution and acetic anhydride. Samples were cleaned up by passing them through the Ludger Clean CEX cartridges. The O-glycans were eluted from the cartridges using water (3 × 0.5 mL). Eluates were dried by vacuum centrifugation prior to fluorescent labelling with procainamide using Ludger procainamide glycan labelling kit (LT-KPROC-24) as described previously[74]. Briefly, samples were incubated with procainamide labelling solution at 65 °C for 1 h. Procainamide labelled glycans were cleaned up using LudgerClean S Cartridges and eluted from the LudgerClean S Cartridges with water (2 × 0.5 mL). The samples were evaporated to dryness under high vacuum using centrifugal evaporation and resuspended in 50 µL water for further analysis. Procainamide labelled samples were analysed by hydrophilic interaction liquid chromatography—fluorescence detection—electrospray ionization tandem mass spectrometry (HILIC-LC-FLR-ESI-MS/MS) using an ACQUITY UPLC® BEH-Glycan 1.7 µm, 2.1 × 150 mm column at 40 °C using a Ultimate 3000 ultra-high-performance liquid chromatography (UHPLC) instrument with a fluorescence detector (λex = 310 nm, λem = 370 nm; Thermo Fisher) controlled by Chromeleon 7.2.2, build 6394. Samples were injected in 20% aqueous/80% acetonitrile; injection volume 25 µL. The following running conditions were used: Buffer A was 50 mM ammonium formate made from Ludger stock buffer # LS-N-BUFFX40; Buffer B was acetonitrile (Acetonitrile 190 far UV/gradient quality; Romil #H049). Gradient conditions were: 0–10 min, 15% A at a flow rate of 0.4 mL/min; 10–75 min, 15–35% A, at 0.4 mL/min; 75–80 min, 35–90% A from 0.4 to 0.2 mL/min; 80–81 min, 90% A at 0.2 mL/min; 81–82 min, 90–15% A at 0.2 mL/min; 82–85 min, 15% A at 0.2 mL/min; 85–95 min, 15% A from 0.2 to 0.4 mL/min. The UHPLC system was coupled on-line to an AmaZon Speed ETD electrospray mass spectrometer (Bruker Daltonics, Bremen, Germany) with the following settings: source temperature 250 °C; gas flow 10 L/min; Capillary voltage 4500 V; ICC target 200,000; maximum accumulation time 50 ms; rolling average 2; number of precursors ions selected 3, release after 1.0 min; Positive ion mode; Scan mode: enhanced resolution; mass range scanned, 200-1600; Target mass, 900.

**Analysis of twitching motility**. *T. thermophilus* HB27 strains were grown at 68 °C for 3 days and at 58 °C for 7 days under humid conditions on minimal medium agar plates[10] containing 0.1% bovine serum albumin. Plates were then stained with Coomassie blue and cells washed off to reveal twitching zones. The colours have been inverted in the images.

**Natural transformation**. *T. thermophilus* wild-type, *pilA4::kat* and *TT_C1836::kat* (*pilA5::kat*) mutants were cultured in TM⁺ media containing appropriate antibiotics for 24 h at 68 °C, 150 rpm (New Brunswick Innova 42, Eppendorf, Hamburg, Germany). These cultures were used to inoculate 10 ml TM⁺ media (with appropriate antibiotics) to a starting OD600 = 0.2 and incubated until OD600 = 0.5 was reached. Thirty microlitre of the cultures were transferred into 370 µl prewarmed TM⁺ medium, and 10 µg genomic DNA from a spontaneous streptomycin-resistant HB27 mutant was added (HB27 Strep). The cultures were incubated for 30 min at 68 °C, 150 rpm. These were subsequently diluted (400 µl in 3 ml TM⁺) and incubated for an additional 3 h at 68 °C, 150 rpm. Transformation samples were plated in suitable dilutions onto TM⁺ agar plates containing 100 µg/ml streptomycin to determine the number of transformants. The number of viable cells was determined by plating on TM⁺ agar plates lacking streptomycin. Following incubation for 2 days at 68 °C, colonies were counted. The transformation efficiency was calculated as the number of transformants per number of living cells.

**Statistics and reproducibility**. For quantitative experiments (Fig. 6b and Supplementary Fig. 2a), n = 4. Gels and blots (Fig. 5c, d, Supplementary Fig. 2b, Supplementary Fig. 4a–c) were all n = 3 (one gel was used for subsequent MS in Supplementary Fig. 2b) and in all cases similar findings were reported. Twitching experiments (Fig. 6a), n = 3. Electron microscopy data (Supplementary Fig. 3) are representative from >100 images and the findings are described by quantitative

data (Supplementary Fig. 3b, c), n = 3. Electron microscopy data (Supplementary Fig. 5) are representative from >1000 images and the findings are described by quantitative data (Supplementary Fig. 5e), n = 3. In Fig. 5e, n = 3 for UHPLC and subsequently one of the samples (Pili wt) was used for MS. For tomography data of cells (Fig. 1), n = 35 tomograms; 20 T4P complexes were assigned to wide or narrow pilus groups, representative images are shown. For tomography data of pili (Supplementary Fig. 1), n = 9 tomograms; >50 pili were assigned to wide or narrow pilus groups, representative images are shown. Data shown in Figs. 2, 3, 4, 5a, b, Supplementary Fig. 6 and Supplementary Fig. 9 are based on two independently determined cryoEM maps from a single dataset of 3138 images.

**Reporting summary**. Further information on research design is available in the Nature Research Reporting Summary linked to this article.

## Data availability

EM maps have been deposited in the Electron Microscopy Data Bank (EMDB, https://www.ebi.ac.uk/pdbe/emdb/) with accession codes EMD-10647 (wide pilus, PilA4) and EMD-10648 (narrow pilus, PilA5). Models have been deposited in the Protein Data Bank (PDB, https://www.rcsb.org/) with accession codes 6XXD (PilA4) and 6XXE (PilA5). The MS proteomics data have been deposited to the ProteomeXchange Consortium via the PRIDE[75] partner repository (https://www.ebi.ac.uk/pride/) with dataset identifier PXD017353. The source data underlying Figs. 5c–e and 6, Supplementary Figs. 2, 3b, c, 4a–c, 5e and 6e, f, and Supplementary Table 4 are provided in the Source Data file. Uncropped versions of gels and blots (for Fig. 5c, d, and Supplementary Figs. 2b and 4a–c) and twitching images (for Fig. 6a) are also shown in Supplementary Fig. 10.

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

## Acknowledgements

We thank Werner Kühlbrandt, Deryck Mills, Janet Vonck and Özkan Yildiz for their support at the MPI of Biophysics in Frankfurt, and Carsten Sachse for indispensable assistance using SPRING and feedback on early versions of this manuscript. We thank Mathew McLaren for maintenance of the EM facility in Exeter and we acknowledge access and support of the GW4 Facility for High-Resolution Electron Cryo-Microscopy, funded by the Wellcome Trust (202904/Z/16/Z and 206181/Z/17/Z) and BBSRC (BB/R000484/1). We acknowledge Diamond for access and support of the cryoEM facilities at the UK national electron bio-imaging centre (eBIC), proposal EM18258, funded by the Wellcome Trust, MRC and BBSRC. We thank Kate Heesom (University of Bristol Proteomics Facility), Imke Wüllenweber and Fiona Rupprecht (MPI for Biophysics) for MS experiments, and Maximilianos Kotsias, Jenifer Hendel, Paulina Urbanowicz and Radoslaw P Kozak (Ludger Ltd) for glycoprofiling. We acknowledge the BBSRC (BB/R008639/1), Max-Planck-Society, the University of Exeter and the Deutsche Forschungsgemeinschaft (AV 9/6-2) for funding.

## Author contributions

Major contributions to (i) the conception or design of the study (A.N., M.S., R.S., B.D., B.A., V.A.M.G.) (ii) the acquisition, analysis, or interpretation of the data (A.N., M.S., R.S., K.K., L.K., K.S., J.D.L., B.D., B.A., V.A.M.G.); and (iii) writing of the manuscript (A.N., M.S., B.D., B.A., V.A.M.G.). All authors commented on the manuscript.

## Competing interests

The authors declare no competing interests.
