## [Peer Review File · Nature Communications]

Reviewers' comments:

Reviewer #1 (Remarks to the Author):

The article by Neuhaus et al reports the detailed structure of two distinct type IV pilus filaments from *Thermus thermophilus* strain HB27. The two pili have different diameters and composition, and presumably exert different biological functions. The main achievement of the paper is that the resolution of the cryo-EM map is the highest (3.2 Å and 3.5 Å, respectively), obtained so far for this family of fibers (resolved at 5- 8 Å). New molecular details revealed by this structure concern primarily the atomic contacts in the so-far elusive hydrophobic core of the fiber. On the other hand, functional analysis of the pili is less conclusive. In particular, the authors propose a model of DNA binding to the thick fibers, even though no biological data is presented to support direct binding.

Major points:

1. The authors use cryo-electron tomography to study the thin and thick Tth pili. They claim that the same machinery is used to assemble both pili. Given the low resolution of the tomograms it is difficult to compare complexes at the base of each type of pili in sufficient detail. What other evidence can support this claim? How many type IV filament-related genes are present in the genome? Although in the past the phenotypes of the pilF and pilQ mutants had been described as non-piliated, was this also tested at low temperatures where piliation efficiency increases ?

2. The residue contacts between pilins and intramolecular interactions in the hydrophobic pilus core are described here for the first time. As they provide novel information, data in the current Figure S7 should be presented in the main text. In the cryoEM reconstructions of PilA4 and PilA5 filaments, the intramolecular interaction between E5 and F1 between the N-terminal amine and carboxyl group of E5 is not really observed. Is the reconstruction sufficiently good for this part? The prepilin peptidase of Tth contains the methylase domain, making it likely that the pilins are N-methylated. However, this does not seem to be taken into consideration in the cryoEM reconstructions. Comparing the electron density of pilin N-terminus with the pseudopilin confirmations obtained in molecular dynamics analysis by Santos Moreno et al, (JMB, 2017) could provide the authors with a useful reference for comparison. In this article, evidence is presented for the neutralizing effect of N-methylation and E5 on pilin-membrane interactions.

3. Related to the above, it would be useful to perform top down MS analysis of pilins PilA4 and PilA5. This would allow the authors to better characterize posttranslational modifications in the pilin globular domains, in addition to N-methylation.

4. The fact that pilA4::kan mutants cannot assemble any pili might be due to a polar effect on nearby genes (unlikely since pilA4 is the last gene in the operon), or to the pilin requirement for the stability of the assembly platform. Can the authors test the latter hypothesis by Western blot by testing the levels of PilMNOPQ complex in the pilA4 mutant ? Importantly, the authors should test whether mutations in pilA4 and pilA5 can be complemented in trans. Complementation experiments are essential for this type of genetic analysis.

5. The authors discuss the role of PilA4 in natural transformation without discussing the recent study by Salleh et al mBio 10 e00614-19, which describes two minor pilin genes from the pilA4 cluster and the ComZ, a likely tip pilin that binds DNA. ComZ interacts with one of the minor pilins from this cluster that resemble strongly the minor pseudopilins in the type II secretion system, suggesting their role in assembly initiation. This point should be included in the discussion (line 279).

6. Related to this, there is no evidence in this work to support the model of DNA binding to the thick pili. I suggest that Fig. 6 be removed or at most be presented in supplementary data.

Minor points:

1. Line 281. It is very unlikely that the major pilin PilA4 forms a capping structure that plays a role in assembly initiation. Structural data presented here argue against this model. However, it might be required for the stabilization of the machinery, as in *Myxococcus* T4P (Chang et al 2016) where deletion of pilin genes leads to the loss of density in cryoET corresponding to the assembly platform proteins.

2. Line 302-308. A proposal in the discussion taken from another paper cannot evidence for this model. Ellison et al in *Nat Microbiol.* 3:773-8; 2018 show evidence that DNA is captured via the tip of the pilus in *Vibrio* and that its binding along the fiber length may not be required for competence. They also show that pilus retraction is not essential for transformation.

Reviewer #2 (Remarks to the Author):

This manuscript by Neuhaus et al. presents a structural study of *Thermus thermophilus* type 4 pili (T4P). Using bacterial genetics, mass spectrometry, cryoET and single-particle cryoEM approaches, the authors identified two different type of T4P present in *T. thermophilus*, wide and narrow. The narrow pilus is enriched in so far uncharacterized pilin that authors named PilA5. The structure of the PilA5 shows the same general features of PilA4, but in a slightly different arrangement. Their study also showed that PilA4 is crucial for natural transformation, while Pil5A is necessary for twitching motility. The identification and near-atomic resolution structural analysis of two forms of T4P from one bacterium for which functions seem to be so clearly separated is an exciting development in the field.

Major comments:

It is intriguing that *T. thermophilus* appear to make two distinct, specialized T4P structures, one responsible for complementation and the other for twitching motility. The authors identify some key characteristics of each that likely contribute to these mechanisms. The paper would be stronger if the authors use this information to experiment rather than speculate. For instance:

In the PilA4/wide/complementation T4P, there is an electrostatic patch of surface-exposed residues that the authors propose binds DNA and provides the rationale for the model in Fig. 6. By performing mutagenesis (e.g., charge reversal of the residues predicted to bind DNA), combined with functional analysis of complementation, the authors could clearly show if this was the DNA binding region of this T4P or not.

In the PilA5/narrow/twitching T4P, there is an appearance of greater flexibility, relative to the PilA4 T4P, which is presumably an advantage for its role in twitching motility. However, are the pili actually more flexible or do they just appear so? Some type of bending/breaking assay of each type of T4P could be informative here.

Both PilA4 and PilA5 T4P appear to be decorated by O-linked glycans. Mutating these serine residues to some other non-Ser/non-Thr residues should make these densities disappear, while lectin binding or glycan array assays would indicate which saccharide moieties are actually present.

Minor comments:

The first half of the Summary is too general and the second half too vague. It would be more informative to describe, in brief, the key findings and the take home message.

There is a general lack of information on the PilA5 sequence, such as, is it shorter compared to PilA4 or longer? What is the sequence identity between these two proteins? It would be helpful to have a sequence alignment of PilA4 and PilA5.

It is very difficult to see pili in FigS3a, middle panel.

We thank the reviewers for their constructive feedback and have responded to their points below. Please note that line numbers stated refer to the tracked changes word document.

Reviewer #1

Major points:

1. The authors use cryo-electron tomography to study the thin and thick Tth pili. They claim that the same machinery is used to assemble both pili. Given the low resolution of the tomograms it is difficult to compare complexes at the base of each type of pili in sufficient detail. What other evidence can support this claim? How many type IV filament-related genes are present in the genome?

There is only one secretin (PilQ) and one assembly platform (PilM, PilN, PilO) present in the *Thermus* genome (Friedrich *et al* (2002), *App Env Micro* 68, 745- 755). In that study, a whole genome search for PilMNOWQ homologues was carried out and only one copy of each of the genes was found. Mutant studies provided clear evidence that the genes are essential for both transformation and piliation. Therefore, the core components of the machinery that assemble wide and thin pili are the same. However, we do acknowledge that there could be differences in accessory protein composition between complexes assembling the two pilus forms. We have therefore clarified our meaning by changing the text to say that the same core machinery is used to assemble both filaments (lines 65-96), and also provided further explanation to this (lines 110-114, 214-219).

Although in the past the phenotypes of the pilF and pilQ mutants had been described as non-piliated, was this also tested at low temperatures where piliation efficiency increases?

PilF is the pilus assembly ATPase and PilQ forms the membrane pore for pilus extrusion, so deleting either of these proteins would mean that no pili would be able to be assembled or extruded, independent of temperature. Nevertheless, we do have data showing that the pilF mutant is impaired in piliation at both temperatures, published in Salzer *et al* (2014), *J Biol Chem*, 289(44): 30343–30354. To clarify, we stated that “mutants defective in the PilQ secretin channel do not extrude pili”. They are however able to assemble them (as we have visualised by cryoET), but cannot push them across the membrane and they accumulate in the periplasm. We have therefore clarified our meaning by re-wording (lines 214-219) and have included the reference for the pilF mutant.

2. The residue contacts between pilins and intramolecular interactions in the hydrophobic pilus core are described here for the first time. As they provide novel information, data in the current Figure S7 should be presented in the main text.

We have moved Fig. S7 to the main text, new Fig. 4. We have also added additional panels to show all intra- and inter-molecular contacts.

In the cryoEM reconstructions of PilA4 and PilA5 filaments, the intramolecular interaction between E5 and F1 between the N-terminal amine and carboxyl group of E5 is not really observed. Is the reconstruction sufficiently good for this part?

Both filaments contain a conserved Glu5 (E5), which in other structures has been modelled with a salt bridge connecting to either the N-terminus of the neighbouring pilin subunit, or to the N-terminus of the same subunit. This is now described in more detail with appropriate references (lines 286-303). We concur that the resolution of our maps is not sufficient to definitively show an obvious salt bridge at the N-terminus. Therefore, we measured the distance between the N-terminus (Phe1/F1) and all potentially negatively charged residues in both models (new Table S3). For both filaments, Glu5 in the same subunit is the nearest negatively charged residue to the N-terminus, hence most likely forms an intramolecular salt bridge (shown in new Fig. 4c,d).

The prepilin peptidase of Tth contains the methylase domain, making it likely that the pilins are N-methylated. However, this does not seem to be taken into consideration in the cryoEM reconstructions. Comparing the electron density of pilin N-terminus with the pseudopilin confirmations obtained in molecular dynamics analysis by Santos Moreno et al, JMB, 2017) could provide the authors with a useful reference for comparison. In this article, evidence is presented for the neutralizing effect of N-methylation and E5 on pilin-membrane interactions.

We agree that there is evidence for N-methylation of pilins and have now taken this into consideration as requested (lines 289-303), including the Santos-Moreno paper. This new section in the Results also relates this point to the previous one regarding E5 and F1.

3. Related to the above, it would be useful to perform top down MS analysis of pilins PilA4 and PilA5. This would allow the authors to better characterize posttranslational modifications in the pilin globular domains, in addition to N-methylation.

We had already performed MS analysis on both PilA4 and PilA5 (Fig. S2). N-methylation could be detected for some, but not all bands that correspond to the proteins (data not shown). Therefore, we have additional proof that pilins can be methylated, but no concrete evidence that F1 is always methylated in our structures. In addition, previous MS analysis (Nivaskumar *et al* (2016), Mol Micro, **101(6)**, 924-941) showed that up to 30% of subunits may remain unmethylated in assembled fibres. We have therefore not included the methylation on F1 in our models (as is also the case for all other T4P structures). We have however taken the neutralising effect into consideration by including the intramolecular salt bridge between F1 and E5 as described in the two previous points.

Regarding additional post-translational modification, we observe extra and unaccounted for density on serine residues only. Based on comparison to other structures and the knowledge that pili are glycosylated, we predicted that densities observed on these serines are O-linked sugars. To confirm this, we have performed additional experiments (new Fig. 5c, which shows gel-based staining of carbohydrates, new Fig. 5d which shows lectin-based binding assays, and new Fig. 5e and Fig. S8 which show the results of detailed glycoprofiling experiments). These confirm that the pilins are indeed modified by sugars and we identify the two most abundant that likely account for the

electron density in our maps. Of interest, we also identify two completely unknown carbohydrate structures that have not been seen before in bacteria.

4. The fact that *pilA4::kan* mutants cannot assemble any pili might be due to a polar effect on nearby genes (unlikely since *pilA4* is the last gene in the operon), or to the pilin requirement for the stability of the assembly platform. Can the authors test the latter hypothesis by Western blot by testing the levels of PilMNOPQ complex in the *pilA4* mutant ? Importantly, the authors should test whether mutations in *pilA4* and *pilA5* can be complemented in trans. Complementation experiments are essential for this type of genetic analysis.

We have investigated both queries as follows:

1). If there are any polar effects on nearby genes in the *pilA4::kan* mutant.

Firstly, we would like to mention that we have seen a failure of complementation *in trans* in the past when trying to complement other mutants of the T4P machinery. This is probably because the amounts of protein produced from plasmids are unbalanced compared to the protein levels expressed from the genome. Nevertheless, complementation plasmids were generated and transformed into the two mutants with the aim of checking for restoration of twitching defects. Unfortunately and as expected, neither could be complemented by providing the wild type *pilA4* or *pilA5* genes *in trans*. Therefore, we devised a different approach to check for the presence of mRNA from genes located downstream of *pilA4* and *pilA5*. We performed PCR using cDNA of wild-type and mutant cells. All transcripts were detected at comparable amounts (new Fig. S4a, b) and demonstrate that there are no polar effects on nearby genes in the *pilA4::kan* (or *pilA5::kan*, also tested for completion) mutants.

2). the pilin requirement for the stability of the assembly platform..by testing the levels of PilMNOPQ complex..

It should be mentioned that a PilMNOPQ complex has not actually been detected in *T. thermophilus* so far. Therefore, we assessed the level of PilQ protein that was able to assemble into multimeric complexes in the membrane. The results are shown in new Fig. S4c and demonstrate that both mutants are unaffected in their ability to assemble membrane embedded PilQ complexes. This shows that the pore is still present in all strains, and the lack of pili observed in the *pilA4::km* mutant is therefore not due to the lack of a secretin channel. It is however still possible that PilA4 is required for the stability of the assembly platform, or the entire multimeric machinery. Being as we do not yet know all of the functions of PilA4 (and had in fact already mentioned additional roles for the protein), we have improved the discussion by including the suggestion that the pilin could also be required for stability (lines 421-425).

5. The authors discuss the role of PilA4 in natural transformation without discussing the recent study by Salleh et al mBio 10 e00614-19, which describes two minor pilin genes from the *pilA4* cluster and the ComZ, a likely tip pilin that binds DNA. ComZ interacts with one of the minor pilins from this cluster that resemble strongly the minor pseudopilins in the type II secretion system,

suggesting their role in assembly initiation. This point should be included in the discussion (line 279).

We have included this paper and improved the discussion surrounding the role of PilA4 and DNA binding (lines 469-482).

6. Related to this, there is no evidence in this work to support the model of DNA binding to the thick pili. I suggest that Fig. 6 be removed or at most be presented in supplementary data.

As this model is only a suggestion of how DNA could bind, we agree with the reviewer and have moved the panel to the supplementary (new supplementary Fig. S9). We feel that it is still worth keeping in the manuscript as it provides a starting point for future mutagenesis and functional experiments.

Minor points:

1. Line 281. It is very unlikely that the major pilin PilA4 forms a capping structure that plays a role in assembly initiation. Structural data presented here argue against this model. However, it might be required for the stabilization of the machinery, as in *Myxococcus T4P* (Chang et al 2016) where deletion of pilin genes leads to the loss of density in cryoET corresponding to the assembly platform proteins.

We have removed mention of PilA4 being involved in the capping structure and have included an additional reference to Chang *et al*, who suggest that the minor pilins form an assembly/priming complex (lines 423-425).

2. Line 302-308. A proposal in the discussion taken from another paper cannot evidence for this model. Ellison et al in *Nat Microbiol.* 3:773-8; 2018 show evidence that DNA is captured via the tip of the pilus in *Vibrio* and that its binding along the fiber length may not be required for competence. They also show that pilus retraction is not essential for transformation.

As per point 5, we have improved the discussion surrounding the role of PilA4 in DNA uptake (lines 469-482), and have included the above paper.

Reviewer #2

Major comments:

1. In the PilA4/wide/complementation T4P, there is an electrostatic patch of surface-exposed residues that the authors propose binds DNA and provides the rationale for the model in Fig. 6. By performing mutagenesis (e.g., charge reversal of the residues predicted to bind DNA), combined with functional analysis of complementation, the authors could clearly show if this was the DNA binding region of this T4P or not.

We have now moved our model of DNA binding to the supplementary data section (suggestion from Reviewer 1) and have improved our discussion on how the filaments might be involved in DNA uptake (lines 469-482). Bearing in mind that we now put much less emphasis on DNA binding to the wide pilus in our manuscript, we believe that performing the time-consuming mutagenesis of the large patch of residues, combined with functional analysis, would not be within the scope of this paper. Our model in Fig. S9 will provide a starting point for future mutagenesis and functional experiments to fully elucidate the mechanism of DNA uptake.

2. In the PilA5/narrow/twitching T4P, there is an appearance of greater flexibility, relative to the PilA4 T4P, which is presumably an advantage for its role in twitching motility. However, are the pili actually more flexible or do they just appear so? Some type of bending/breaking assay of each type of T4P could be informative here.

We have performed detailed statistical analysis of filament curvature, and this is displayed in a new Fig. S5e. This shows that the narrow PilA5 pili are more curved than the wider PilA4 counterparts, and thus are likely to be more flexible. We do however appreciate that curvature is not necessarily the same as flexibility. [Redacted] We have therefore changed the description in the manuscript to say that the narrow filaments appear to be more flexible (based on our analysis of curvature shown here), rather than say that they are definitively so.

3. Both PilA4 and PilA5 T4P appear to be decorated by O-linked glycans. Mutating these serine residues to some other non-Ser/non-Thr residues should make these densities disappear, while lectin binding or glycan array assays would indicate which saccharide moieties are actually present.

We have performed a gel-based fluorescent staining experiment to show that both PilA4 and PilA5 are indeed coated by sugars (new Fig. 5c). To investigate further, we have also performed lectin-based binding experiments as the reviewer suggested, and this shows that both proteins are positive for GalNAc, mannose and/or glucose (new Fig. 5d). We have also performed detailed glycoprofiling experiments (new Fig. 5e, Table S4) which identify the two most abundant sugars that likely account for the electron density in our maps. We also identify two completely unknown carbohydrate structures that have not been seen before in bacteria. These data clearly demonstrate that both of our pili are decorated by O-linked glycans. To perform the experiments that the reviewer describes would require multiple rounds of mutagenesis, followed by clarification that cells are still viable, pili

assembled and functional, purification of filaments, cryo data collection and processing of large data sets to determine two new structures without the sugars present. As this is a huge amount of work and we do not think we will gain significant information from it, we did not think it was a good investment of resources, especially considering the new data that we have included with respect to glycosylation.

Minor comments:

4. The first half of the Summary is too general and the second half too vague. It would be more informative to describe, in brief, the key findings and the take home message.

We have re-worded the Summary as suggested.

5. There is a general lack of information on the PilA5 sequence, such as, is it shorter compared to PilA4 or longer? What is the sequence identity between these two proteins? It would be helpful to have a sequence alignment of PilA4 and PilA5.

We had already included a sequence alignment in Fig. S6 (now Fig. S7), which must have been overlooked. It is referred to on line 243.

6. It is very difficult to see pili in FigS3a, middle panel.

We have changed this figure to one where the pili are easier to see.

REVIEWERS' COMMENTS:

Reviewer #1 (Remarks to the Author):

The authors have addressed all my comments in the revised version.

Reviewer #2 (Remarks to the Author):

I am satisfied with the revisions that the authors have made to the manuscript and support its publication in its current state.